# Mitigating Forgetting in Continually Pretraining MoE-LLMs by Adding and Chilling Experts

## Abstract

As model training requires more and more compute, the cost of re-training models to support new data or domains increases as well. Methods to adapt existing models to new data distributions are crucial to avoid spending redundant compute re-training models from scratch. However, naïve finetuning often incurs forgetting of previously learned capabilities. In this paper, we analyse how different factors such as model size, dataset size and replay data impact forgetting when adapting models to new data distributions. We also propose to increase the capacity of Mixture-of-experts models by adding new experts and reducing the learning rate of the old model weights. Our experiments show that this simple method allows one to reduce forgetting and learn efficiently on the new domain.

## 1 Introduction

Pretraining models uses a lot of compute, a cost that continues to increase as models become better and larger. Today, very few organizations have the resources to train their own frontier language models from scratch. A few general-purpose large language models (LLMs) have been made publicly available. These models are incredibly strong on general English understanding tasks, but fail on other languages or more targeted domains that haven't been considered when assembling the pretraining corpus. As new use cases are identified for LLMs, organizations keep spending compute to train new models from scratch. Reusing previously trained models to adapt them to new domains or languages is crucial to save time and compute.

However, naïvely finetuning models on a new data distribution can easily lead to forgetting: while learning new capabilities, some of the existing ones are lost. Continual Learning research focuses on training models on a stream of data whose distribution changes over time. Commonly used techniques focus on modularity and the use of samples from previous data distributions, called replay data. Continual Pretraining (CPT) (Gupta et al., 2023; Ke et al., 2023) tackles adaptation to new data distributions at a larger data scale, similar to a pretraining setting. In this paper, we focus on the pretraining of large Mixture-of-Experts (MoE) transformer models (Shazeer et al., 2017; Lepikhin et al., 2020; Fedus et al., 2022) on a sequence of datasets with different distributions. This setting is inspired by realistic use cases such as adapting a large publicly available model to handle new languages, or updating an already deployed model to incorporate new skills, knowledge, or domains. We study how model size, dataset size, and data shift impact forgetting of these models when switching to new data distributions. Additionally, we propose to add new experts to MoE models and to reduce the learning-rate of already trained experts, which we call chilling. By training mostly the new experts, we show that our method Add-and-Chill-Experts (ACE) mitigates forgetting while increasing performance on the new domain. Using replay-data is a common way to reduce forgetting, but leads to lower performance on the new domain on a fixed budget. ACE allows one to use less replay, and get better performance on the new domain. Similarly to our work, Chen et al. (2023) completely freezes the old experts and requires using output-level regularization to obtain good performance. We show that choosing the right learning-rate for different parts of the model is a simple yet effective way to mitigate forgetting across a wide range of replay-amount.

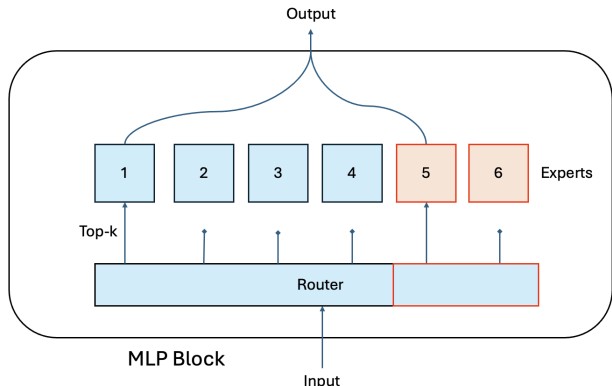

Figure 1: **Diagram explaining our method Add-and-Chill Experts.** Added weights have a red border. The router and the old experts shown in blue have a reduced learning rate compared to the new experts.

## 2    RELATED WORK

**Language Models**    Deep neural networks have shown strong performance on many language tasks, especially since the advent of pretrained transformer models (Vaswani, 2017; Devlin, 2018; Radford et al., 2019). Many works have demonstrated the power of increasing model and dataset sizes through empirical scaling-laws (Hestness et al., 2017; Kaplan et al., 2020; Hoffmann et al., 2022). Training larger models on larger datasets has proven the quickest way to obtain stronger models. In this race for compute, sparse architectures like MoEs provide an efficient way to train larger models while reducing the amount of compute needed (Jacobs et al., 1991; Shazeer et al., 2017). Recent works have combined MoE architectures with transformers to train impressive models (Lepikhin et al., 2020; Fedus et al., 2022; Du et al., 2022). Many strong publicly available LLMs rely on MoE transformer architectures (Dai et al., 2024; Jiang et al., 2024; Muennighoff et al., 2024; Yang et al., 2024).

**Continual Learning for NLP**    Continual Learning research explored using replay (Robins, 1995; Shin et al., 2017; Lopez-Paz & Ranzato, 2017), modular architectures (Rusu et al., 2016; Yoon et al., 2017; Wen et al., 2020; Mallya & Lazebnik, 2018), and regularization (Kirkpatrick et al., 2017). These methods have recently been applied to NLP problems (Wu et al., 2022). These works often consider task-specific adaption of models, with relatively small datasets. Gupta et al. (2023) studies the impact of learning-rate and warm-up in continual pretraining. Instead of considering task-specific and small to medium-size datasets, continual pretraining focuses on new domains with larger datasets. The data scale is more similar to those used in LLM pretraining. Ke et al. (2023) uses a soft-masking mechanism for domain adaption. Chen et al. (2023) is most relevant to our work. They propose to add new experts to MoE transformer models to effectively learn new tasks without too much forgetting. Their work shows that freezing the old experts together with output-level regularization are necessary to achieve good results. We remove the need for regularization which requires additional compute, and instead propose a simple method using adequate learning rates.

## 3    ADD-AND-CHILL EXPERTS

We propose to use the modularity of the MoE transformer architecture to mitigate forgetting. When switching to a new domain, we add experts and we freeze or chill (reduce the learning rate of) the old weights to preserve the previously learnt capabilities, as presented in Figure 1.

The new experts are initialized by copying weights from existing experts. The new router weights are initialized to zero. The number of experts-per-token is kept the same. This means that the compute-per-token remains unchanged, despite the new model having more parameters.

Table 1: Hyper-parameters of pretrained models

| Model | 8x50M | 8x200M | 8x800M |
|---|---|---|---|
| batch size | 1M | 1M | 2M |
| learning rate | $1e{-}3$ | $3e{-}4$ | $3e{-}4$ |
| hidden | 512 | 1024 | 1536 |
| # attention heads | 8 | 8 | 12 |
| # head groups | 2 | 2 | 4 |
| # experts | 8 | 8 | 8 |
| # experts per token | 2 | 2 | 2 |
| # layers | 8 | 12 | 22 |

## 4 EXPERIMENTAL SETUP

Our experiments are performed with three sizes of MoE transformer models: 8x50M, 8x200M and 8x800M parameters. Each model goes through two phases of training: pretraining, and continued-pretraining (CPT). We use a standard procedure for pretraining, and focus our experiments on the CPT phase. The goal is to learn on the new domain while retaining capabilities acquired in pretraining. Finally, in the CPT phase we also experiment with Mixtral (Jiang et al., 2024), a publicly available model, to provide a more realistic use-case.

### 4.1 DATASETS

The datasets used in the two phases are drastically different, thus inducing forgetting when naïvely continuing training in the second phase. We switch from general-purpose web-crawled English data to source code from GitHub. Other plausible scenarios could be: adapting an English model to other languages, or adapting a general-purpose model to conversational and agentic data.

**Pretraining** In the pretraining phase we use Fineweb (Penedo et al., 2024), a large dataset of high-quality text data curated from CommonCrawl. We will refer to this as the upstream domain or pretraining domain.

**CPT** In the CPT phase, we use The Stack v2 (Kocetkov et al., 2022; Lozhkov et al., 2024), a large corpus of source code from GitHub. We refer to this as the downstream domain or CPT-domain. We consider a scenario where some replay-data (Fineweb) can be included in CPT to mitigate forgetting. The drawback of adding replay-data is that compute is spent on the pretraining-domain instead of the CPT-domain, and thus lowers the performance on the CPT-domain compared to a run without replay-data. We consider data-mixes with 0%, 10%, 50%, 90%, and 100% replay-data.

### 4.2 METHODS AND BASELINES

All the models used in this paper are MoE transformer models (Vaswani, 2017; Shazeer et al., 2017), using Rotary positional embeddings (Su et al., 2024) with tied word embeddings. All the experiments use sequence-length 2048, except the experiments using Mixtral that use 8192 and untied word embeddings. We use the Adam (Kingma, 2014) optimizer with weight-decay 0.1, $\beta_1 = 0.9$ and $\beta_2 = 0.95$. The models are trained with a load-balancing loss coefficient of 0.02, except for CPT with added experts where we use 0.006 instead. Assuming that there is some degree of specialization of old experts on the upstream domain and of new experts on the downstream domain, our intuition was that enforcing a uniform usage of experts might hurt the performance or de-specialize the experts. However in a short series of experiment we found negligible impact of modifying the coefficient of the load-balancing loss. Table 1 shows the hyper-parameters of each model.

### 4.2.1 PRETRAINING

Following the learnings from Hägele et al. (2024) we pretrain models with a linear warmup, fixed learning rate, and minus-squareroot cooldown from intermediate checkpoints. We obtain pretrained models with variable numbers of pretraining tokens, ranging from 120B to 600B. We perform short ablations on 20B tokens to select the best learning-rate for each model.

### 4.2.2 CONTINUED PRETRAINING

We compare 3 different protocols for the second phase of training:

**CPT - vanilla**   The baseline model is naïvely trained on the new domain for 100B tokens. We use a linear warmup and constant learning-rate. The learning-rate was selected from an ablation on 20B tokens of 100% replay. We found that continuing training on the pretraining-domain by re-warming the learning rate to a slightly larger value (e.g. $3e{-}5$ instead of $1e{-}5$ for the 8x200M model) induces forgetting. Selecting a small enough learning rate was crucial to mitigate forgetting, even in a setting with only replay-data.

**Added Experts - vanilla (AE)**   We add new experts to the model following the procedure described in Section 3. This variant is trained with an equal learning-rate for all parts of the model, following the same schedule as the vanilla-CPT baseline.

**Add and Chill Experts (ACE)**    In this variant, we add new experts and chill the rest of the model. The new experts are trained with a higher learning-rate, while the rest of the model uses a learning-rate 10x smaller. In later experiments, we explore other ratios of learning-rates between new and old experts to find better configurations.

### 4.3 EVALUATION METRICS

We evaluate models on their pretraining-domain performance and CPT-domain performance using the loss on the respective datasets: Fineweb and The Stack. We also use Hellaswag (Zellers et al., 2019)[1] to measure natural language commonsense, a typical capability that would improve by training on datasets like Fineweb (pretraining). We use HumanEval (Chen et al., 2021)[2] to measure coding capabilities. Training on The Stack (CPT) should improve the model's performance on this benchmark.

## 5 EXPERIMENTAL RESULTS AND ANALYSIS

### 5.1 MAIN EXPERIMENTS

**More pretraining makes a model more prone to forgetting.**   For the three model sizes, we perform CPT with 50%-replay starting with checkpoints after cooldown with various amounts of pretraining tokens. Note that in this experiment with did not search for the best learning-rate, but chose to use a learning rate 10x smaller than the one used in pretraining for each model. We see a significant regression in upstream loss after CPT, as shown in Figure 2. The more tokens the base model has been trained on, the worse is the forgetting. As an example, the baseline-CPT that started from the 8x200M checkpoint at 500B tokens regresses to a loss of $2.40$, which is equivalent to the value of the loss at 250B tokens: we throw-away half of the pretraining-compute during CPT, even though 50% of the CPT-compute was dedicated to replay. There is a similar trend for the larger and smaller models: the more the model has been pretrained, the more prone to forgetting it is.

**Larger models are less prone to forgetting.**   Figure 2 also shows that forgetting seems much less severe on the large model than it is for the two smaller models. The 8x800M model after 250B pretraining tokens maintains the same upstream performance after CPT, although forgetting does occur for longer pretraining.

---

[1]We use the likelihood-based implementation from EuleutherAI's lm-evaluation-harness
[2]We use the implementation from bigcode-evaluation-harness

Table 2: Performance of 8x200M MoE on the pretraining-domain Hellaswag (accuracy) and the CPT-domain HumanEval (pass@1) after CPT with 50% replay. All models use the same amount of compute due to sparsity. We indicate the learning-rate used for the new experts, old experts, and the rest of the model, including all the router weights. * marks the methods that appear in Figure 3

| Model | # Experts | Learning-rate ($\times 10^5$) | | | Hellaswag | HumanEval |
|---|---|---|---|---|---|---|
| | | new | old | rest | | |
| Pretrained* | 8 | – | – | – | **44.9** | 0 |
| Baseline* | 8 | – | 1 | 1 | 44.3 | 4.0 |
| Baseline | 8 | – | 3 | 3 | 43.6 | 6.7 |
| AE - vanilla* | 10 | 1 | 1 | 1 | 44.2 | 3.4 |
| AE - vanilla | 10 | 3 | 3 | 3 | 43.7 | 7.0 |
| ACE - freeze | 10 | 10 | 0 | 0 | **44.9** | 0 |
| ACE - chill more | 10 | 10 | 0.1 | 0.1 | **45.0** | 7.9 |
| ACE* | 10 | 10 | 1 | 1 | 44.4 | 6.3 |
| ACE +4 experts | 12 | 10 | 1 | 1 | 44.6 | **9.3** |

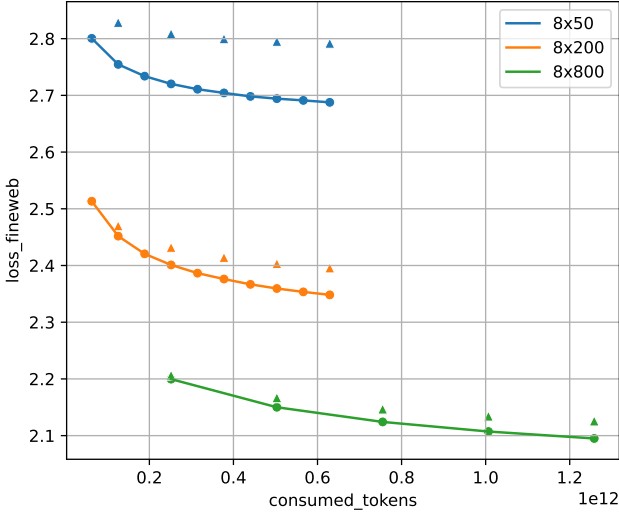

Figure 2: **Increase of loss after CPT.** The curves show the upstream loss of pretrained checkpoints after cooldown. The triangles show the value of the loss after CPT for select checkpoints. The position of triangles on the horizontal axis is given by the number pertraining tokens, not taking into account the tokens from CPT. All CPT used 100B tokens and 50% replay-data.

**Adding replay-data reduces forgetting, at the cost of downstream performance.** We select the 8x200M checkpoint trained for 500B tokens to compare different CPT methods described in Section 4.2. We perform CPT starting with this checkpoint with various data-mixes that incorporate from 0% to 100% replay-data. Using replay-data allows one to maintain performance on the pretraining-domain, at the cost of downstream performance. Figure 3 shows this trade-off for different CPT methods. All the curves clearly show that replay is a crucial factor in controlling the trade-off between upstream and downstream performance. By using more replay we prevent forgetting, but also significantly hurt downstream performance. As shown on Figure 3b, the baseline CPT maintains a good performance on Hellaswag when using at least 90% of replay, but has close to 0 performance on HumanEval. Using less than 50% replay leads to catastrophic forgetting.

**Adding experts can mitigate forgetting** Figure 3 shows that the trade-off offered by AE is not better than baseline CPT. This indicates that adding capacity alone does not mitigate forgetting. Our method ACE shows superior performance overall, with the best trade-off between Hellaswag and HumanEval performance. Choosing appropriate learning-rates for different parts of the model is key

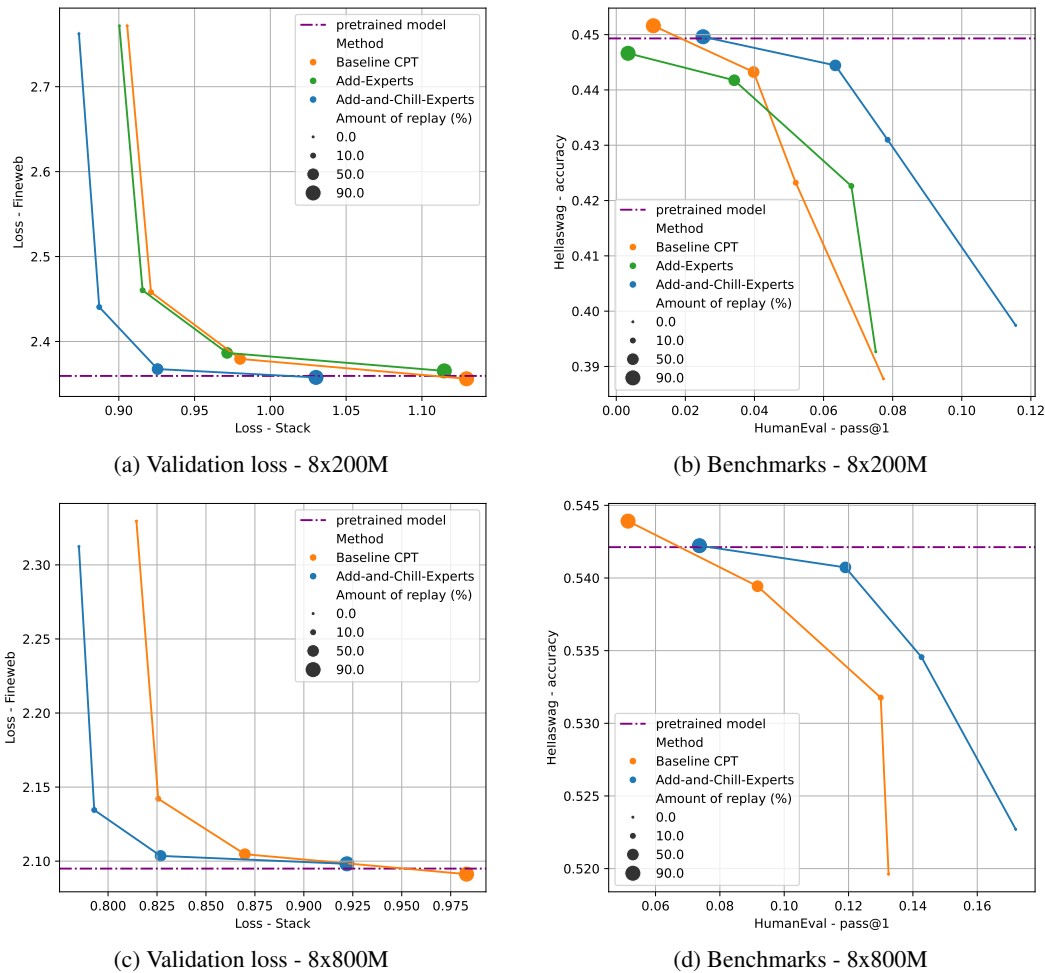

(a) Validation loss - 8x200M

(b) Benchmarks - 8x200M

(c) Validation loss - 8x800M

(d) Benchmarks - 8x800M

Figure 3: **Upstream and downstream performance trade-off.** Each line shows the performance trade-off for a given method, with varying amount of replay. The size of the dots reflects the amount of replay: 0, 10, 50 or 90%. (left) Performance measured with validation loss on Fineweb (pre-training) and the Stack (CPT). (right) Performance measured with benchmarks: Hellaswag and HumanEval.

to retaining upstream performance while learning the new domain. The 8x200M ACE model that used 50% replay preserved most of the Hellaswag performance at 44.4% accuracy, while performing well on HumanEval with 6.3% pass@1. Figures 3c and 3d show that the findings also hold for the larger 8x800M model, with ACE providing a better trade-off overall.

**Further exploration.** Previous experiments added 2 new experts, and used a 10x larger learning-rate for new experts for ACE. We explore a few more hyper-parameter settings, keeping the same 8x200M base checkpoint pretrained for 500B tokens. The results are shown in Table 2. As seen previously, ACE gives good performance on both domains. *ACE - freeze* and *ACE - chill more* show that it's possible to retain all of the base model's capabilities by chilling the model more. Freezing the rest of the model completely prevents the model from learning on the new domain, but *ACE - chill more*, with a ratio of 100 between old and new expert learning rate, shows the best performance trade-off. Finally, by adding 4 new experts instead of 2, we maintain the same level of performance on Hellaswag and bring HumanEval performance to 9.3, which is the best of the models in this experiment. These experiments show that even better trade-offs are possible by searching the space of hyper-parameters offered by our method.

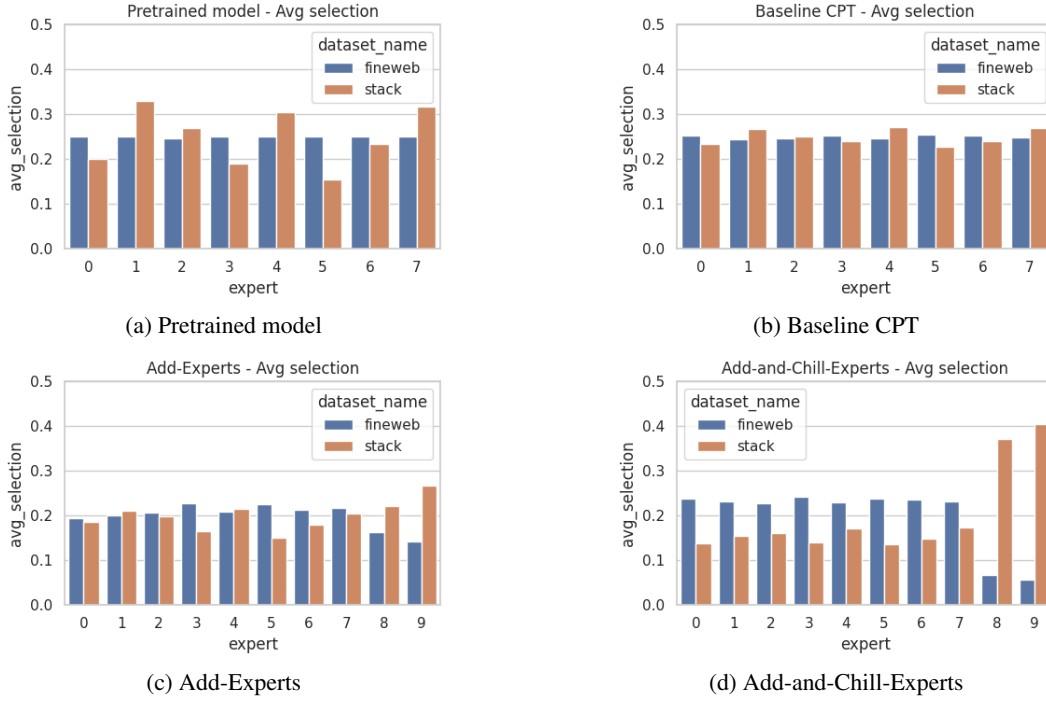

Figure 4: **Average expert selection of the 8x200M models on the pretraining and CPT domains.** The expert selection is averaged over all layers. For (c) and (d) the expert indices 8 and 9 correspond to the new experts.

## 5.2 EXPERT SPECIALIZATION

To better understand the different methods, we examine the impact of CPT methods on the selection of experts depending on the domain. We consider the 8x200M models trained with 50% replay data. Based on 40 random samples from Fineweb and the Stack, we compute the expert selection averaged over tokens and layers. The histograms of average selection are shown in Figure 4. The base model 4a has uniform selection on Fineweb, enforced by the load-balancing loss. The expert selection on the Stack is not uniform as the model has not been trained on this domain. The baseline CPT 4b shows uniform selection for both domains. AE 4c shows a slight specialization of experts, as the new experts (indices 8 and 9) tend to be more selected on the CPT domain and less on the pretraining domain. ACE 4d specializes the experts even more.

## 5.3 REALISTIC USE-CASE

In this experiment we apply our method to the publicly available Mixtral model (Jiang et al., 2024) to enhance its coding abilities. This is a realistic use-case where users may want to leverage publicly available models and improve some of its capabilities in a specific domain like coding. As with many publicly available models, the pretraining dataset is not released so using replay-data is not an option. Catastrophic forgetting is a common issue in that scenario, although it might be mitigated by the large size of the model. We perform CPT on the Stack starting with the base Mixtral model with a learning-rate of $3e-6$ and a batch-size of 2M tokens. For ACE, we use hyper-parameters based on the best performing model in Table 2: we add 4 new experts, the new experts use a 10x larger learning-rate. Due to limited compute resources, we limit the CPT phase to 60B tokens. Results are shown in Table 3. Interestingly, results do not show improvement from using ACE in this setting. ACE performs worse than the baseline on Hellaswag and Humaneval by a small margin. The baseline does not show any sign of forgetting to begin with, preserving the performance on Hellaswag, and even slightly improving compared to the base model. In this scenario where no forgetting is observed with vanilla CPT, there is no benefit in adding more experts. Based on our results presented in Figure 2, it is likely that Mixtral was relatively under-trained compared to other

Table 3: Performance of Mixtral 8x7B on Hellaswag (accuracy) and HumanEval (pass@1) after CPT on the Stack. All models use the same amount of compute due to sparsity. We indicate the learning-rate used for the new experts, old experts, and the rest of the model, including all the router weights.

| Model | # Experts | Learning-rate ($\times 10^6$) | | | Hellaswag | HumanEval |
|---|---|---|---|---|---|---|
| | | new | old | rest | | |
| Pretrained | 8 | – | – | – | 64.9 | 38.3 |
| Baseline | 8 | – | 3 | 3 | 65.5 | 41.7 |
| ACE | 12 | 30 | 3 | 3 | 64.5 | 40.9 |

models used in our experiments. We speculate that base models pretrained on more tokens would be more prone to forgetting. A longer CPT phase might also induce more forgetting.

## 6 CONCLUSION

We performed experiments that clearly show the impact of model size, dataset size, and replay-data on forgetting during CPT. We proposed ACE, a simple method that consists in adding experts to a pretrained MoE transformer model, and chilling the old experts and the non-expert weights. Our experiments show that this simple method is effective at improving the performance on the new domain while retaining all the capabilities of the pretrained model. Some additional experiments using more experts and different ratios of learning-rate showed that even better trade-offs are possible. Experiments with Mixtral showed no benefit of using our method in a scenario where baseline CPT does not induce forgetting. Future work will explore expanding dense models to Mixture-of-experts with ACE.

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
