# OpenReview forum: "Mitigating Forgetting in Continually Pretraining MoE-LLMs by Adding and Chilling Experts"
_ICLR.cc/2025/Conference — Submitted to ICLR 2025_

### Official Review · Reviewer_DQ6w · 2024-10-28

**Soundness:** 2
**Presentation:** 2
**Contribution:** 1
**Rating:** 3
**Confidence:** 4

**Summary:**

This paper introduces MoE to the continual pretraining setting and proposes ACE (Add-and-Chill-Experts) to mitigate forgetting while increasing performance on new domains. ACE expands Mixture-of-Experts models by incorporating new experts while reducing the learning rate for the existing model weights. Experimental results demonstrate that the effectiveness of this straightforward method.

**Strengths:**

1.	The paper writing is clear and the method is easy to follow.

2.	The experimental analysis is consistent with previous work and is intuitive.

**Weaknesses:**

1.	The method is too simple, and the description is not clear. There is no explanation of how to design routers and experts, and there is no in-depth analysis of the method's advantages and disadvantages. It is like a technical report rather than an academic paper.

2.	Lack of innovation. There are many similar works in MoE+CL [1,2,3], and the conclusions are well-known. For example, adding experts can reduce forgetting (e.g., prompt-based CL), and replay data will reduce forgetting while reducing the performance of new tasks (stability-plasticity balance).

3.	The review of related work is not comprehensive, as many papers on Mixture-of-Experts (MoE) combined with continual learning (CL) are not cited. Additionally, the understanding of existing CL methods appears to be insufficiently thorough.

4.	The experiment lacks other comparison methods, and the experimental effect is not significantly improved.


[1] Boosting continual learning of vision-language models via mixture-of-experts adapters. CVPR, 2024.

[2] Mixture of Experts Meets Prompt-Based Continual Learning. arXiv, 2024.

[3] Theory on Mixture-of-Experts in Continual Learning. arXiv, 2024.

**Questions:**

Please refer to weaknesses.

---

### Official Review · Reviewer_Hktb · 2024-11-03

**Soundness:** 3
**Presentation:** 3
**Contribution:** 2
**Rating:** 5
**Confidence:** 3

**Summary:**

In this paper, authors propose ACE for CPT. ACE consists of two process: (1)adding new experts (2) and chilling the old ones. In addition, authors investigate the factors that impact forgetting, including (1) pretrained data size; (2) model size; (3) replay data size; (4) expert numbers.  Experiments are conducted on Fineweb for pretrained domain and Stack v2 fro CPT domain.

**Strengths:**

This paper is well organized. Analyzing the factors that impact LLM forgetting is valuable, providing insights to the CPT field of LLM.

**Weaknesses:**

1) Limited novelty: The strategy of freezing old experts and allocating new experts for new tasks is somewhat simplistic, leading to limited methodological innovation in this paper.
2) Lack of Comparison with Chen et al. (2023): As the authors claim, "Chen et al. (2023) is most relevant to our work." and "We remove the need for regularization which requires additional compute, and instead propose a simple method using adequate learning rates." The authors should provide a comparison with Chen et al. (2023) in terms of performance and training efficiency to align with their claims. These crucial comparisons are missing from the paper.
3) Limited Datasets Used in Experiments: The paper only uses two datasets, fineweb and stack. Validation on more datasets is necessary.
4) Lack of Settings for ACE (new, old, rest=10, 0, 0.1) in Table 2: This setting is missing, which may fail to demonstrate that chilling old experts is a better choice than freezing old experts.
5) The results in Section 5.3 (REALISTIC USE-CASE) are interesting, but they also indicate that the ACE method is not very robust for different pre-training LLMs.

**Questions:**

1) Why does using more replay data reduce the performance on new task, or even prevent learning the new task? The authors state, "As shown in Figure 3b, the baseline CPT maintains good performance on Hellaswag when using at least 90% of replay, but has close to 0 performance on HumanEval." In my view, the more replay data that is used, the closer the resulting model should be to the solution of joint training, which should yield good performance on both tasks.
2) The authors claim that "More pretraining makes a model more prone to forgetting." In my view, better upstream pretraining should provide the model with stronger generalization abilities, making it easier to adapt to new tasks and reducing the forgetting. The conclusion presented here seems to contradict this. Could this be because there is a significant difference between the pretraining domain and the CPT domain? If forgetting is related to the pretraining data, then the claim that "More pretraining makes a model more prone to forgetting" may not always hold true.

---

### Official Review · Reviewer_bfir · 2024-11-03

**Soundness:** 3
**Presentation:** 3
**Contribution:** 1
**Rating:** 3
**Confidence:** 3

**Summary:**

This work indroduce Add-and-Chill Experts (ACE) method for the continual pretraining (CPT) of LLM Mixture-of-Experts (MoE) with exemplars. ACE is continual learning (CL) method base on the architecture expansion - the new experts are added to the mixture for the new task. To prevent forgetting of the old knowledge better - "the chilling" part is proposed - lower lr is applied for the existing network parameters. The authors evaluated ACE on two datasets, representing domain shift from general text (Fineweb dataset) to code (The Stack).

**Strengths:**

1. Simple method for CPT and MoE LLM modes.
2. Paper easy to follow.
3. Experiment with Mistral model.

**Weaknesses:**

1. Limited experimental section and insights. Feels like a very small number of experiments were done. Even the one that are provided, sometimes are questionable, e.g. Tab. 2 - ACE -freeze with rest = 0, how the model could train the routing for the new task? For me it's more like a sanity check. What I'd like to see is the ACE - freeze 10/10/0/{0.1,1.0} - so adding new experts and freezing the old model, but allow training the routing(rest).

2. The setting is limited to 2 datasets and we taking the results only from the one domain shift: generic text -> source code, with two different evaluations. I find it very limited if we're considering continual model pre-training.

3. Limitations are not discussed. We're adding new experts - increasing the number of parameters with each task. Maybe some analysis of the compression or other techniques could be considered and make this work better. I know that the authors motivated this by compute-per-token, but those values are not provided at the end in the experimental section to compare.

4. Experiment with Mistral raises some issues and presents more like the opinions of the authors (line: XXX It is likely ... relatively... we speculate... a longer CPT phase might... ). This experiments only raises more question if the provided results so far are generalizable to other settings (?).

I find this work very interesting. However, in the current shape it feels still like work in progress, with some limited experiments and outcomes that are not well understood/analyzed.

**Questions:**

1. Are the results from a single run experiments?
2. Tab 2. - What are the results for ACE - freeze 10/10/0/{0.1,1.0} and ACE*/ACE+4 experts and 10/0/1?
3. Why only two datasets? If the compute is the problem - why not trying smaller LLMs and datasets?

---

### Meta-Review · Area_Chair_UyKC · 2024-12-21

**Metareview:**

The paper studies how different factors (e.g., model size, dataset size and replay data) contribute to forgetting when adapting
models to new data distributions. It proposes to increase the capacity of mixture-of-experts models by adding new experts and reducing the learning rate of the old model weights. Experiments have been conducted to assess the effectiveness of the proposed approach.

**Strengths**
- It is important to understand how relevant factors may affect the forgetting issue when adapting pre-trained model to new tasks from different data distributions.

*Weaknesses**

- Reviewers share common concerns, including limited technical novelty, lack of discussion/comparison with relevant existing models, and insufficient evaluation.

Since no rebuttal is provided, none of the major concerns above has been addressed, leading to the rejection recommendation of the paper.

**Additional Comments On Reviewer Discussion:**

The authors did not provide any rebuttal so no further discussions were conducted.

---

### Decision · Program_Chairs · 2025-01-22

Reject